# Dynamics of Primary Social Networks to Support Mothers, Fathers, or Guardians of Transgender Children and Adolescents: A Systematic Review

**DOI:** 10.3390/ijerph19137941

**Published:** 2022-06-28

**Authors:** Paula Daniella de Abreu, Rubia Laine de Paula Andrade, Israel Lucas da Silva Maza, Mariana Gaspar Botelho Funari de Faria, Jordana de Almeida Nogueira, Aline Aparecida Monroe

**Affiliations:** 1Ribeirao Preto College of Nursing, University of Sao Paulo, Ribeirao Preto 14040-902, Brazil; pauladdabreu@usp.br (P.D.d.A.); israelucas2008@gmail.com (I.L.d.S.M.); rimagod@usp.br (M.G.B.F.d.F.); amonroe@eerp.usp.br (A.A.M.); 2Clinical Nursing Department, Federal University of Paraiba, Joao Pessoa 58059-900, Brazil; jalnogueira31@gmail.com

**Keywords:** parents, transgender persons, social support, social networking

## Abstract

Mothers, fathers, or guardians of children and adolescents who do not identify with the gender they were assigned at birth face barriers in their social network to recognize their children’s gender identity. This study aimed to analyze the scientific evidence on the dynamics of primary social networks to support mothers, fathers, or guardians of transgender children and adolescents. This is a systematic review of qualitative studies guided by the PRISMA guidelines. Controlled and free vocabulary were used to survey the studies in the EMBASE, Scopus, MEDLINE, Cumulative Index to Nursing and Allied Health Literature (CINAHL), PsycInfo, Latin American and Caribbean Literature in Health Sciences (LILACS), and Web of Science databases. A total of 21 studies composed the final sample. Primary social networks were described as fragile and conflicting family/blood relationship ties with disapproval, isolation, rejection, a lack of understanding, and feelings of exclusion were expressed. Some have lost friends, reported tension in marriage and with relatives, and were commonly treated with hostility and harassment. Social transition does take place in the mutual context of struggle and resistance which demands a support network for parents or guardians.

## 1. Introduction

In the course of growth and development, some children and adolescents spontaneously experience their gender identity, which may be incompatible with what they were assigned at birth [1,2].

A cisnormative social pattern, which defines gender based on biological sex, can make it difficult for transgender (trans) children and adolescents to recognize their gender identity and manage their needs, especially when facing secondary sexual characteristics and transphobic situations [3,4,5,6].

In some studies, mothers, fathers, or guardians of transgender children and adolescents commonly voice doubts, fears, anxiety, and concern for the future and safety of their children. In addition to this, they experience isolation and grief related to an ambiguous loss [7] of an idealized child, as well as uncertainties and stressors that affect the entire family’s quality of life [8,9,10]. Ambiguous loss is a relational phenomenon, and it assumes an attachment to the missing person [7]. This situation is exacerbated by the lack of transgender representation in social spaces, which also contributes to ignorance, tension, intolerance, and delay in recognizing their children’s gender identity, with the need to face pathologizing stigmas and transphobia [8,9,10].

Social transition to recognize gender identity takes place amidst a relational context, which is driven by social interactions that can provide strengthening of emotional, face-to-face, instrumental, and informational support, as well as self-support, in order to provide changes from a situation of dependence to that of autonomy of the target audience [11,12]. From this perspective, mothers, fathers, or guardians of transgender children and adolescents actively participate in such a transition, i.e., the family “transitions” together [10].

Existing supportive dynamics or controlling conflicting, weakened, or broken relationships can be identified in primary social networks (family ties, blood relationship, friendship, neighborhood, and work), according to the Social Network framework proposed by Sanicola [11]. The present study aimed to analyze the scientific evidence on the dynamics of primary social networks in supporting mothers, fathers, or guardians of transgender children and adolescents. An analysis of scientific evidence can contribute to affirmative actions for trans visibility and endorse the criticality of the theme in the different social and political contexts around the world, mainly because these are vulnerable people whose identities and gender expressions are considered crimes in some countries. The political moment must be considered for an in-depth analysis of genocidal actions veiled in discriminatory speeches from powerful leaders that have repercussions on transphobia, social deaths, and murders which are still recurrent, even in so-called democratic countries.

As recommended by Page et al. [13], the study was registered in the “PROSPERO: A registry for systematic review protocols” platform (registry: CRD42022301747). Furthermore, preliminary searches in databases (MEDLINE, Cochrane Database of Systematic Reviews, JBI Evidence Synthesis and in OSFREGISTRIES) and in the PROSPERO platform indicated the innovative potential of this study in relation to the field of scientific evidence available, since no recorded protocol or review on the theme was identified.

## 2. Materials and Methods

This is a systematic review of qualitative studies guided by the Joanna Briggs Institute Manual for Evidence Synthesis of systematic reviews of qualitative evidence [14] and Preferred Reporting Items for Systematic Review and Meta-Analysis (PRISMA) recommendations [13]. Qualitative research aims at interpreting meanings in the sociocultural field, such as in studies involving families and other social networks, in order to appropriate relevant phenomena, including the field of policies and practices in health [15].

The present study was developed in six stages: theme and guiding question identification; establishment of inclusion and exclusion criteria; definition of information to be extracted from selected studies; assessment of studies included in the systematic review; interpretation of results; and synthesis of knowledge [16].

A guiding question was established in order to reach the proposed objective and follow the steps, namely, “What is the scientific evidence on the dynamics of primary social networks in supporting mothers, fathers, or guardians of transgender children and adolescents?”, defined through the PICo strategy as: P (population: mothers, fathers, or guardians of transgender children and adolescents); I (phenomenon of interest: social support); Co (context: social network). Children and adolescents were defined as people up to 19 years of age, according to the World Health Organization (WHO) classification [17].

In accordance with the Social Network theoretical framework [11], the guiding question listed studies that responded to the investigation on the dynamics of primary networks, with the description of relational phenomena presenting alliances, conflicts, discontinuity, ruptures, wear and tear, and transgressions in networks. Thus, it was possible to identify relationships and functions that compose it.

Additionally, original qualitative articles, in all languages, that dealt with the dynamics of primary social networks through experiences verbalized by mothers, fathers, or guardians of transgender children and/or adolescents were included. Duplicate publications, gray literature (abstracts published in proceedings, newspaper news, dissertations, theses, book chapters, letter to the editor, preprint publications) and studies with results regarding the population of interest which were not presented separately from other populations were excluded.

An article search was carried out in December 2021 through the Central Library system of the University of São Paulo and CAPES journal, which provided access to the following databases: EMBASE, Scopus, MEDLINE, Cumulative Index to Nursing and Allied Health Literature (CINAHL), PsycInfo, Latin American and Caribbean Literature in Health Sciences (LILACS), and Web of Science. No publication year or language limits were used for the searches.

The search strategy consisted of controlled and free vocabulary combined by the Boolean operator OR to distinguish them and the Boolean operator AND to associate them, in order to integrate and direct the maximum number of studies on the theme. The search strategy was adapted to each database according to its specificities—see published protocol [18].

After surveying the studies in the databases, they were transferred to the online Rayyan QCRI platform [19] for exclusion of duplicate studies and subsequent reading of titles and abstracts by two independent researchers, and then a third evaluator for a decision in cases of disagreement or doubt between the first two. The studies selected in this first stage were submitted to full reading, which allowed to analyze their relevance in relation to their inclusion in the review. The study selection process is presented in the flowchart, according to the Preferred Reporting Items for Systematic reviews and Meta Analyses (PRISMA 2020) recommendations [15].

Data extraction was performed using a form adapted from Lockwood et al. [14], composed of the following variables: author; year and journal of publication; phenomenon of interest (objective); method (place of study, participants, data analysis); and main results. The methodological quality assessment of studies included in this review was carried out using a checklist for assessing qualitative research, proposed by The Joanna Briggs Institute [14].

The results were submitted to a narrative synthesis in the light of the Social Network framework proposed by Sanicola [11].

## 3. Results

The study selection is detailed in a flowchart (Figure 1), which indicates the survey of 9447 publications in the databases and the final selection of 21 qualitative articles to compose the study sample.

The articles were published in 2009 (4.7%) [20], 2011 (4.7%) [21], 2013 (4.7%) [22], 2014 (4.7%) [23], 2015 (4.7%) [24], 2016 (9.5%) [25,26], 2018 (4.7%) [27], 2019 (9.5%) [28,29], 2020 (19%) [30,31,32,33], and 2021 (33%) [34,35,36,37,38,39,40]. All articles (100%) were published in English, of which 14 (66.6%) [20,21,23,24,25,26,27,28,30,31,32,35,37,40] were from America, 6 (28.5%) [29,33,34,36,38,39] were from Europe, and one (4.7%) [22] was from Oceania. The information and the synthesis of the main results of the final sample of all articles included in the review are presented in Table 1.

Regarding the methodological quality of the articles included in this review, all showed congruence between the philosophical perspective and research methodology, between proposed research methodology and research question or objectives, and between research methodology and data collection methods. Participants and their voices were adequately represented, fulfilling the registrational issues and conclusions of data analysis or interpretation [20,21,22,23,24,25,26,27,28,29,30,31,32,33,34,35,36,37,38,39,40].

Limitations were identified regarding congruence between research methodology and data representation and analysis which were not clear in one of the studies [20], and were not presented in another [29]. There was inconsistency between research methodology and interpretation of results in one study [40]. There was a lack of clarity in the statement which positions the researcher culturally or theoretically in seven studies [20,21,22,25,27,28,39,40], while it was found that the researcher’s influence on the study was not addressed, and vice versa, in one study [29]. These results are summarized in Table 2.

## 4. Discussion

### 4.1. Recognition of Gender Identity and the Challenges Faced

Recognizing gender identity and social transition of transgender children and adolescents is surrounded by guilt, fears, and uncertainties of mothers, fathers, or guardians, who sometimes experience grief related to the ambiguous loss, psychological loss with the person still physically present [7], and of their idealized children [23,24,26,27,30,33,35]. A study carried out in the United States of America (USA) revealed that most parents recognized their children’s gender identity, some ignored it, and others were not involved in decisions related to social transition [20].

Issues related to gender identity that emerged in the studies included in this review were permeated by parents’ concerns with the social transition of their transgender children, especially with communication, use of pronouns, and clothing consistent with gender identity [27,30,35], in addition to issues related to peer and family support [27,37]. In parallel, their need for self-care was also listed as a prerogative to understand their own limits and how to act with their children [27]. In one of the studies, two parents mentioned that participation in research, with the verbalization of their experiences, constitutes a form of self-reflection for their self-support [38].

Parents feel challenged by their external environment, family, friends, neighbors, or even co-workers [32]. Studies have shown that the decision to support their transgender children in the social transition resulted in the potential loss of friends [21,24,26,27,29,32,35,40]. In other studies, participants also reported being under stress in their marriage and with close relatives [20,21,24,25,33]. One of the studies also mentioned a mother’s disagreements and concern with her child’s biological father for not being “supportive” [37]. The nuclear family tends to expose itself to harassment and hostility from relatives [21,39] and neighbors [39].

The discrimination that parents of transgender children and adolescents face can be seen in the absence of support from peers, parents, and the community, family members and friends. Moreover, there are cases called misgendering, situations in which there is the incorrect use of pronouns for gender identity, which may occur intentionally or inadvertently, configuring itself as another obstacle for parents in the transitional process of their children [29]. In another study, parents stated having been reported to the authorities by other parents [21].

### 4.2. The Existing Ties in the Primary Social Networks

There is a bond of mutual dependence between guardians and their transgender children for decision-making and establishing bonds which can be supportive or conflicting [34]. Sometimes, mothers’, fathers’, or guardians’ needs are not considered; however, they need to be sustained in terms of the support they provide to their children [32], especially as a result of the frequent conflicting social pressures due to children’s gender identity, which generates a feeling of conflict in their private and public lives. In one of the studies, some participants reported feeling uncomfortable discussing the issue with their family, friends, and neighbors [24]. In another study, participants’ cultural notions of gender in the social network relative to friends/neighborhood were inflexible about the gender transition of their children [23].

Family and blood relationship ties proved to be fragile and conflicting in the extended family. Disapproval and rejection have been reported with the isolation of the nuclear family [26,32,33]. Other studies have also referred to rejection and broken ties from adult family members and friends [21,29,35,36]. The dynamics in the nuclear family tend to be weakened by the need for mental health actions and welcoming all family members [28]. In a study carried out in Italy, a mother revealed a weakened/broken bond with her eldest daughter and son-in-law, as well as their fear that their son would be “influenced” by his brother [36].

The lack of information from family members was mentioned as an obstacle to understanding gender identity [21,30], in addition to frustration of parents when trying to prove to family members that their child was transgender [35].

Advocacy was mentioned as being important among family members and co-workers [27]. Another father reported that co-workers use discriminatory expressions, making him mediate the discussion on the topic [38]. Some parents assess the need to disclose the information to co-workers. While some disclose it, others avoid talking openly at work to avoid relational conflicts [36].

One of the studies emphasized that relationships for parents who have a child with gender variation change throughout the family, since there is an impact on the family system with “non-acceptance” by the siblings, including a veiled disapproval from relatives. Furthermore, fathers took longer to accept their children than mothers. Siblings also mentioned the feeling of loss due to the transition and the rest of the family initially had resistance and place blame on the mother [23]. Mothers often receive greater emotional, physical, and organizational burden from caring for their transgender children [30].

Parental need included support from family and friends [22]; however, families of transgender children and adolescents present different stages of understanding and acceptance of their gender identity and gender expression. An important initial step in family therapy would be to assess levels of opposition and/or support in the family, with the aim of strengthening family ties [40].

In a study carried out in the USA, mothers discussed the reactions of their family members, such as aunts, uncles, cousins, and grandparents, in which they expressed hesitation or initial resistance, followed by the recognition of gender identity and agreement with children’s transition [23]. Other studies have also reported conflicts and discouragement on the part of grandparents [20,26]. In another study, friends or family believed that “encouraging” a gender identity could contribute significant harm to children [40].

### 4.3. The Lack of Informational Support and Its Impact

Parents mentioned a lack of informational support [21,22,23,28,29,30,31,36,37,38,39] and search for knowledge on their own, for instance, through research on websites and blogs, or even online contact with transgender people [30,33,36,37,38]. The internet is sometimes a means of disseminating materials that are difficult to understand in terms of language related to gender variation. It comprises materials with information “against gender variation in childhood”, “best not to read”, or “sensationalist articles”. In one study, parents reported that this subject was taboo in the family [38]. In another study, parents reported receiving or wanting support from Facebook groups or other online resources and also from media such as their child’s school, therapy, support groups, friends, peers, and books [37].

Another study refers to possible ways of providing information: books and stories about children with gender variations and their families; up-to-date research published in the media; guidelines and strategies for fatherhood, with these being essential to not only be available online, but in waiting rooms in health services, libraries, and social media programs [22]. Informational support for parents was greater through contact with gay or transgender friends [38].

Parents’ knowledge of gender identity emerged from their own experience of their children [28], transgender adolescents who sometimes become the parents’ teachers [39], and the experiences acquired by parents can also be related to their profession. In one of the studies, the participant was a social worker and received support at the workplace and used her knowledge to apply with her child at home [28].

Support from family and friends in an environment that provides recognition and care, respect, empathy, and encouragement from family and friends allows parents to manage their children’s needs [22].

Thus, having a child with gender variation impacts the entire family system. Parents’ reaction can also impact the reaction of their social network, as in cases where parents do not support their children and siblings, they also tend to not support their transgender children [25]. Fragile family dynamics demand health actions for each family member, with the appreciation of family demands, especially parents [28].

The studies explicitly showed that the existing relationships in primary social networks were anchored in trans invisibility in social spaces. This aspect, added to the historical and political context of delay in affirmative action, has repercussions on the cyclical model of transphobia and the imminent risk of regression of rights mainly conquered through the struggle of organized civil society. Therefore, this study contributes to a critical analysis that goes beyond primary social networks, as it indicates the major role of the academic community in the production of health knowledge which goes beyond the limits of health services and its biological approaches: it bleeds into the social field and its interference in the well-being of a population and maintenance of socially invisible lives. The authors of this review are cisgender nurses and present their perspectives from this point of view, but they agree with strengthening scientific literature produced by trans people about their experiences in their own perspectives.

It is important to emphasize that the methodological quality of the studies included in this review were satisfactory, however it is recommended that identified limitations need to be considered in future studies in order to allow clarity of qualitative studies with greater rigor and detail in the steps developed by the researchers. The presentation of in-depth nuances integrates and instrumentalizes the information necessary for understanding the issue under study.

The present study had limitations, such as the non-inclusion of gray literature and manual search of studies. On the other hand, it considered that original articles published in journals indexed in the chosen databases were sufficient to envision the proposed investigation and prioritize the methodological quality of studies that meet the methodological rigor and quality of the content provided for by experts.

## 5. Conclusions

The dynamics of primary social networks of mothers, fathers, or guardians of transgender children and adolescents mentioned in the studies included in this review revealed conflicts and ruptures in family, blood relationships, friendship, and neighbors ties.

Recognizing gender identity is the critical point in alliances and transgressions, since welcoming behaviors, advocacy, formulation, and execution of policies and rights in the social and health fields are built from there. Children’s or adolescents’ social transition does not occur in isolation, but culminates with family transition in the mutual context of struggle and resistance.

## Figures and Tables

**Figure 1 ijerph-19-07941-f001:**
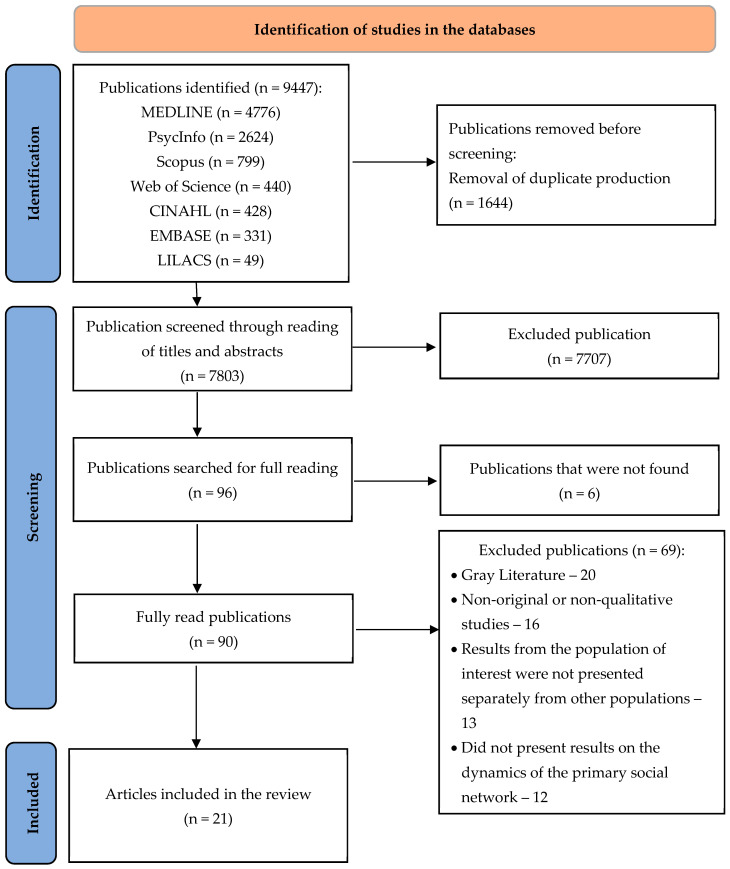
Flowchart of studies selected for systematic review on the dynamics of primary social networks to support mothers, fathers, or guardians of transgender children and adolescents. Ribeirao Preto, SP, Brazil, 2022. LILACS—Latin American and Caribbean Literature on Health Sciences; CINAHL—Cumulative Index to Nursing and Allied Health Literature. Source: Adapted from Page et al. [13].

**Table 1 ijerph-19-07941-t001:** Information and synthesis of the main results of studies selected for systematic review on the dynamics of primary social networks to support mothers, parents, or guardians of transgender children and adolescents. Ribeirao Preto, SP, Brazil, 2022.

ID	Author(s)/Journal/Year/Country	Objective	Study Sites Participants	Collection Data Analysis	Main Results
S1 [20]	Hill; Menvielle/Journal of LGBT Youth/2009/USA	Document problems faced by parents of children with childhood gender variation behaviors and/or gender variation identity and compile their knowledge.	- Wisconsin, Washington District of Columbia and Canada. - 43 parents (heterosexual and lesbian couples) of 31 transgender children.	- Semi-structured open-ended interviews through the phone. - Analysis: not applicable.	Most parents recognized their children’s gender identity, some ignored it, and the others were not involved in decisions related to social transition. One parent reacted quite negatively to this and created conflicts with his wife. Few fathers reported being happy with their children’s gender identity, however, mothers were more “open”, although one of them reported embarrassment. In some families, the dynamics was training and policing of children’s behavior, sometimes due to the insistence of grandparents.
S2 [21]	Riley et al./International Journal of Sexual Health/2011/USA	Provide a foundation to support all children with gender variation and their parents by identifying their needs.	- USA, Australia, Canada, and the United Kingdom. - 27 mothers; 3 parents and 1 guardian of transgender children; with snowball sampling.	- Interviews via internet with closed-ended and open-ended questions. - Content analysis, with the help of Weft qualitative data analysis (QDA), with a continuous reflective-interpretative process to generate the themes.	Most parents reported a sense of exclusion, with some writing in detail about how they lost friends and relatives by affinity. They also reported tension in the marriage or how the family is harassed, ostracized, and/or treated with hostility. Some parents were reported to the authorities by another parent who claimed that a child’s variant gender behavior meant she was being abused at home.
S3 [22]	Riley et al./Sex Education/2013/Australia	Investigate and understand the experience of people who have the experience and knowledge necessary to determine the needs of children with gender variations and their parents.	- Sydney (Australia). - Parents of children 12 years of age and younger, transgender adults, and clinical professionals with experience working with the transgender community, with snowball sampling.	- Online interview with closed and open questions, by the Zoomerang method. - Grounded theory and content/thematic analysis, which involved a reflection on the interpretative process. - A coding method was used, following Buckingham and Saunders.	The needs of parents of children with gender variation include information (could be provided in the form of books and stories about children with gender variation and their families; up-to-date research published in the media; and parenting guidelines and strategies); and support from family and friends (an environment of acceptance and care, with respect, compassion, help, and encouragement from family and friends, allows parents to manage the needs of their children).
S4 [23]	Kuvalanka et al./Journal of GLBT Family Studies/2014/USA	Understand how parents of transgender children come to identify their children’s expressions of diverse gender, how they feel about their children’s expression and also understand how their social contexts impacted the family’s experiences.	- Miami, Oxford, Ohio, (USA). - 5 mothers of transsexual children aged between 8 and 11 years recruited through social networks and contacts of professionals of the study’s advisory board.	- Phone interviews. - Open codification (inductive thematic analysis), with three main thematic fields emerging.	Four mothers discussed the reactions of their relatives, such as aunts, uncles, cousins, and grandparents. Most of the time, extended families expressed hesitation or initial resistance, but eventually accepted children’s transitions. One participant noted that connecting with other parents of transgender children provided the kind of support she could not receive from her friends.
S5 [24]	Sansfaçon et al./Journal of LGBT Youth/2015/Canada	Understand the problems and challenges experienced by parents of children with gender variation in the process of supporting their children’s gender identity and expression as they grow up.	- Montreal, Canada. - 14 parents of children with gender variance.	- Participatory action research using social action methodology (SAM) principles and processes and focus groups. - Grounded theory. Data were analyzed as they were collected and involved open, axial, and selective coding.	Parents are constantly pressured by social reactions to their children’s gender identity and sometimes felt as if their private and public lives were in conflict. Some felt uncomfortable discussing it with their neighbors, friends, and family. It was reported: “You risk losing your family, marriage, network”. Family members in some cases showed strong resistance.
S6 [25]	Grey et al./Family Process Institute/2016/USA	Describe the experience of parenting of a child with gender variation as well as the mutual influence between the child, the family, and the environment.	- Boston, Massachusetts (USA). - 11 caregivers, 8 mothers, and 3 fathers.	- Interviews with semi-structured questionnaire. - Participatory action research using social action methodology (SAM) principles and processes and focus groups.	All parents described how having a child with a gender variation impacts the entire family system. The marital relationship between parents oscillated between agreement and tension. Most reported that their partners agreed with them on the shared goals of parents for both “rescue” and “acceptance” of gender expression, although one or the other was further ahead. These approaches impacted the relationship between transgender children’s siblings. Parents who were “deniers” described “mockery” by the siblings, and those who “accepted” reported “acceptance” and defense by the siblings. In the extended family, there was rejection and possible “acceptance” followed by implicit disapprovals.
S7 [26]	Pyne/Journal of Progressive Human Services/2016/Canada	Focus a lens on parents of transgender children who affirm their children’s sense of gender, explore how these parents know their children’s gender identities and develop a theory to better understand the knowledge underlying the decision to assert children’s self-identities.	- Canada. - 15 parents of gender-non-conforming children and adolescents up to 12 years old.	- Semi-structured interviews. - Grounded theory, performed in three coding steps, open, axial, and focused or selective, to incorporate the categories.	Participants reported experiences of conflict with grandparents and other family members as well as judgment by other parents. In some cases, parents witnessed their child(ren) being rejected and harassed by other children. When parents decided to affirm the identity of their children, as all participants eventually did, many were judged along with their children. There were reports of isolation of family and friends after supporting their children’s transition.
S8 [27]	Alegría/International Journal of Transgenderism/2018/USA	Understand the experience of parents and caregivers of transgender children/adolescents and their relationships with close family members.	- Six American states (four on the west coast and two on the east). - 14 fathers/mothers/guardians.	- Semi-structured interviews. - Inductive analysis using a comparative method. - Emerging themes were identified.	Relationship of mutual dependence between parents and children. “Need to tell”: assessment of disclosure to their social network. Advocacy between family members and co-workers. Family transition with possible changes of friends. Self-support: knowing their limits.
S10 [28]	Carlile/International Journal of Transgenderism/2019/United Kingdom	Investigate the experiences of transgender children and adolescents and their families in their interactions with primary and secondary healthcare providers in England.	- England, the United Kingdom. - 65 transgender and non-binary parents and children between 12 and 18 years old and other adults.	- Participant-researcher model, here called “Illuminate”. - Analysis with identification of themes and sub-themes, outlined and evidenced in a document.	Dynamics in the nuclear family weakened by the need for actions aimed at parents and siblings’ mental health; importance of family support for recognizing gender identity. Parents’ knowledge of gender identity came from their own experience with their children and experiences related to their profession. One of the participants was a social worker and received support in the workplace, also used her social work skills to perform a reflective “daily work” with her child at home.
S11 [29]	Hidalgo; Chen/Journal of Family/2019/USA	Explore how, if at all, parents of prepubescent transgender people experience gender minority stress related to their children’s gender identity/expression.	- USA. - 40 parents, 8 participating alone and 16 participating as dyad and their children: 24 children and adolescents aged between 4 and 11 years treated in a gender clinic.	- Focus groups, following a scripted protocol. - Content analysis. A multiphase coding process to establish reliability was employed. - Dedoose program was used.	Parents reported perceiving discrimination from other parents and unknown community members (lack of support from peers with cisgender children), in addition to being “examined” and “observed”. They also reported rejection perceived or experienced by family members and friends (adults) and other parents, with “silence”, judgments, broken ties with friends and other parents in activities segregated by gender. In addition, there are cases of misgendering (use of intentional or inadvertent incorrect pronouns) and fragile relationships: family-based non-affirmation, lack of support from some family members that may become another obstacle.
S9 [30]	Sansfaçon et al./Journal of Family Issues/2019/Canada	Explore the journey of parents of transgender children regarding the acceptance of their children’s gender identity, including the reactions to their children’s transformations, struggles, facilitators of acceptance and experiences in clinical settings.	- Montreal, Quebec; Ottawa, Ontario and Winnipeg, Manitoba. - 4 fathers and 32 mothers of 35 children and adolescents from 9 to 17 years old.	- Semi-structured interviews. - Inductive and reflective thematic analysis. Transcripts were coded and separated into thematic areas. - The data analysis software used was MAXQDA.	The mother often carries the entire emotional, physical, and organizational burden. While the lack of co-parent involvement and support did not appear to prevent the parents we interviewed from giving their children access to gender affirmation and transition-related care, it was said that this delayed the process. Some parents also mentioned that not having prior knowledge and information about transgender identities and issues was a definite barrier in their process of understanding and recognizing their children’s transgender identity.
S13 [31]	Clark et al./Elsevier Journal of Adolescence/2020/Canada	Exploring the decision of transgender youth and their parents made about the start of hormone therapy.	- British Columbia (Canada) - 21 transgender youth aged between 14 and 18 years and 15 parents of these young people - Snowball sampling technique was used.	- Semi-structured interviews. - Analysis of constructivist grounded theory of transcriptions of interviews and drawings of lifelines, performed within the groups, then between groups, with the help of NVIVO 11 Pro.	The processes of recognition of their children’s gender identity took place after their children’s revelation that they requested parental support for hormone therapy. The majority supported their children’s access to hormone therapy in order to meet their urgent demand. However, parents felt overwhelmed and lacked informational support, and, after meeting their children’s demand, they returned to the discovery phase to try to understand the situation and their role.
S14 [32]	Medico et al./Clinical Child Psychology and Psychiatry/2020/Switzerland	Examines the experiences of transgender and gender-diverse children and youth and their parents/caregivers who have been referred to gender-affirming clinics.	- Switzerland. - 10 parents/caregivers and 10 children, one child, eight adolescents and one young individual. - Snowball sampling was used.	- Semi-structured interviews. - Inductive analysis through reading, line-by-line coding and the help of MAXQDA software, and then organized into themes.	Parents reported being challenged by their external environment, family, friends, neighbors or even co-workers. As a mother said, “I have received many comments”, and her decision to support her child in a process of transition and/or identity affirmation as transgender is criticized.
S15 [33]	Testoni; Pinducciu/Sciendo/2020/Italy	Consider how parents of transgender children handled their transition and how they live the experience of grief.	- Italy, Spain and USA. - 18 parents (11 cis-females, 6 cis-males, 1 non-binary). Spain (5), Italy (6), and USA (7).	- Individual online interviews, through the SurveyMonkey platform. - Thematic analysis, with the help of Atlas.ti.	Parents expressed their social isolation and the difficulty of being understood about their own suffering and experience of loss in their family network and social relationships. Difficulties in social acceptance, family and couple support can prolong and complicate the grief phase of depression and sadness. Parents showed that they felt excluded from social contexts and that communication with family members and networked relationships were often very difficult. Within family relationships, the most painful was with the intimate partner, in the parental couple role. Parents were often confronted with the exclusion of social contexts and, above all, with the rejection of partners or family members.
S12 [34]	Bhattacharyaet al./Journal of Family Psychology/2020/USA	Understand the perspectives of transgender youth and their caregivers, young-caregiver relationship and caregiver–caregiver in the family system.	- USA. - 20 families (20 transgender youth aged between 7 and 18 years and 34 caregivers).	- Semi-structured interviews - Thematic analysis with immersion and crystallization method - Dedoose program was used.	Strong bond of dependence with support/conflict between young caregivers. Relationship between conflicting parents and divergent ideas: mother’s recognition and father’s adjustment period. The mother found support in friendships and the father did not disclose it to friends (refers to shame and difficulty). Closeness between couples contributed to support for the young person. With regard to conflict, one caregiver blamed the other. There was conflicting and interrupted ties in the extended family with an episode of transphobia.
S16 [35]	Dangaltchevaet al./Frontiers in Psychology/2021/Canada	Describe the adaptation of the Connect program to meet the needs of parents of transgender and gender-non-conforming youth and measure program effectiveness.	- British Columbia, Canada. - 20 parents (14 mothers and 6 fathers) of 16 gender-non-conforming youth aged between 12 and 18 years.	- Group dynamics; participation in 9 out of 10 sessions. - Model analysis, through notes of recorded sessions, reviewed and assigned themes, using NVivo 11.	Gender identity disclosure by their children generated a “shock” in their parents, others perceived signs previously, but they did not understand. There were reports of parents’ frustration trying to prove to family members that their son was trans. Other parents mentioned ties that were broken with family and friends. They had no friends to talk to about the transition. The relationship between parents and children for the use of names and pronouns was referred to as a challenge, and “not loving the transition” weakened the relationship with their children.
S17 [36]	Frigerio et al./Journal of GLBT Family Studies/2021/Italy	Explore the experiences of parents of transgender adolescents diagnosed with gender dysphoria who, for the first time, attended a clinic for psychological consultation.	- Milan (Italy). - 15 parents (10 mothers and 5 fathers) of transgender and gender-diverse adolescents, most (93%) transgender boys, aged between 14 and 19 years.	- Individual interviews, via Skype or phone. - Inductive thematic analysis through coding, search of themes, and organization of themes.	Some family members and close friends supported parents to prioritize the well-being of their transgender children, other family members were less familiar with transgender children due to geographic distance or work and relational obligations, non-recognition, “acceptance” was frequent. A mother reported a fragile/interrupted bond with her eldest daughter and son-in-law for not recognizing and fearing that her son would be “influenced” by her brother. Some parents disclosed their child’s gender identity so as not to have to be justified, including in the workplace. Others avoided speaking openly to relatives and friends of some and hiding it from co-workers to avoid relational conflicts.
S18 [37]	Katz-Wise et al./Journal of Family Issues/2021/USA	Explore attitudes and challenges faced by parents/caregivers of transgender and/or non-binary youth.	- USA. - 27 parents/caregivers of transgender and/or non-binary children, adolescents, and young people.	- Interview through online form. - Thematic analysis, using immersion/crystallization approaches to identify themes. Primary coding was completed using the Dedoose software platform.	Social interactions within the family: Sub-theme 4 included the disclosure of child gender identity to the father and the family’s level of focus on the child’s gender identity. It also included aspects of support and relationships between parents, children, and extended family. In general, participants expressed support for their children’s gender identity and gender identity revelations. One mother reported disagreement with her child’s other father: “His biological father is not supportive and I am worried...”. Participants who currently received or desired support reported receiving or wanting support including from friends and colleagues.
S19 [38]	Lorusso; Albanesi/Journal of Community & Applied Social Psychology/2021/Italy	Map/describe the needs of parents of transgender and gender-diverse children in Italy, their relationship with the health and education systems, and how they deal with the challenges of the context in which they live.	- Italy. - 13 parents recruited by the snowball technique. Cisgender parents = four fathers and nine mothers, among them: three heteroparentals and one female homoparental. The age of their children ranged from 5–17 years.	- Individual semi-structured interviews via Skype. - Reflective thematic content analysis, in which the first themes were generated, these were validated by participants by email for building the final themes.	Parents reported that they had no information on gender variation in children and adolescents prior to the experience with their children. This subject was a taboo in the family. One father mentioned having explained to friends when they asked about his transgender daughter with her registered name, another father reported that co-workers use discriminatory expressions, which led him to mediate a dialogue. Two parents mentioned participating in interviews as a form of self-reflection for their self-support.
S20 [39]	Rabain/Frontiers in Sociology/2021/France	For parents: Reflect on family relationships and deal with all types of experiences of discrimination.	- Paris (France). - Parents and adolescents in support groups, together between two and twenty families.	- Group interviews with progressive inclusion. - Identification of recurring themes arising from group therapeutic approaches for parents and adolescents and for both (multifamily).	The most recurrent issue among participants included hostility from relatives outside the family nucleus and neighbors. Most of the time, this phenomenon of rejection constitutes an impediment to the transition process. With regard to informational support, transgender adolescents temporarily become teachers of their own parents. The latter not only have subjective knowledge, but also information collected on social networks and blogs by transgender adolescents. Collective discourse is based on a lexical field of neologisms that they have to convey to their parents.
S21 [40]	Szilagyi; Olezeskib/Smith College Studies in Social Work/2021/USA	Discuss unique challenges encountered in working with parents and caregivers of transgender youth during virtual visits that have the potential to interfere with the development of a therapeutic alliance and the movement towards greater family acceptance.	- Connecticut (USA). - Parents/caregivers of transgender adolescents participated.	- Meetings by videoconference of team members with children and guardian together then separately. - Description of two clinical cases for interpretation and discussion.	The relationship with close friends and family was judgmental by some if parents supported gender variation and by others if they did not. Friends or family believed that “encouraging” an identity could lead to significant damage to their child. Among their politically progressive friends were several adults who identified as gay, lesbian or queer, some of whom had atypical or non-conforming gender expressions, and all of whom “accepted and affirmed”. Transgender youth families have different stages of understanding and accepting gender identity and gender expression of their young people, an important initial step in family therapy would be to assess levels of opposition and/or support in the family, with the aim of increasing family attunement.

**Table 2 ijerph-19-07941-t002:** Methodological quality of articles included in the systematic review on the dynamics of primary social networks in supporting mothers, fathers, or guardians of transgender children and adolescents. Ribeirao Preto, SP, Brazil, 2022.

	1. Is There Congruence between the Stated Philosophical Perspective and the Research Methodology?	2. Is There Congruence between the Research Methodology and the Research Question or Objectives?	3. Is There Congruence between the Research Methodology and the Methods Used to Collect the Data?	4. Is There Congruence between the Research Methodology and the Representation and Analysis of Data?	5. Is There Congruence between the Research Methodology and the Interpretation of Results?	6. Is There a Statement Locating the Researcher Culturally or Theoretically?	7. Is the Researcher’s Influence on Research and Vice versa Addressed?	8. Are Participants and Their Voices Adequately Represented?	9. Is the Research Ethical According to Current Criteria or, for Recent Studies, Is There Evidence of Ethical Approval by an Appropriate Body?	10. Do the Conclusions Drawn in the Research Report Stem from Data Analysis or Interpretation?
Hill; Menvielle, 2009 [20]	Y	Y	Y	U	Y	U	Y	Y	Y	Y
Riley et al., 2011 [21]	Y	Y	Y	Y	Y	U	Y	Y	Y	Y
Riley et al., 2013 [22]	Y	Y	Y	Y	Y	U	Y	Y	Y	Y
Kuvalanka et al., 2014 [23]	Y	Y	Y	Y	Y	Y	Y	Y	Y	Y
Sansfaçon et al., 2015 [24]	Y	Y	Y	Y	Y	Y	Y	Y	Y	Y
Grey et al., 2016 [25]	Y	Y	Y	Y	Y	U	Y	Y	Y	Y
Pyne, 2016 [26]	Y	Y	Y	Y	Y	Y	Y	Y	Y	Y
Alegría, 2018 [27]	Y	Y	Y	Y	Y	U	Y	Y	Y	Y
Carlile, 2019 [28]	Y	Y	Y	N	Y	U	Y	Y	Y	Y
Hidalgo; Chen, 2019 [29]	Y	Y	Y	Y	Y	Y	N	Y	Y	Y
Sansfaçon et al., 2019 [30]	Y	Y	Y	Y	Y	Y	Y	Y	Y	Y
Clark et al., 2020 [31]	Y	Y	Y	Y	Y	Y	Y	Y	Y	Y
Medico et al. 2020 [32]	Y	Y	Y	Y	Y	Y	Y	Y	Y	Y
Testoni; Pinducciu, 2020 [33]	Y	Y	Y	Y	Y	Y	Y	Y	Y	Y
Bhattacharya et al., 2020 [34]	Y	Y	Y	Y	Y	Y	Y	Y	Y	Y
Dangaltcheva et al., 2021 [35]	Y	Y	Y	Y	Y	Y	Y	Y	Y	Y
Frigerio et al., 2021 [36]	Y	Y	Y	Y	Y	Y	Y	Y	Y	Y
Katz-Wise et al., 2021 [37]	Y	Y	Y	Y	Y	Y	Y	Y	Y	Y
Lorusso; Albanesi, 2021 [38]	Y	Y	Y	Y	Y	Y	Y	Y	Y	Y
Rabain, 2021 [39]	Y	Y	Y	Y	Y	U	Y	Y	Y	Y
Szilagyi; Olezeskib, 2021 [40]	Y	Y	Y	Y	U	U	Y	Y	Y	Y

Legend: Y—Yes; U—Unclear; N—No.

## Data Availability

Data could be found in Table 1.

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
