# Peer review of "Dynamics of Primary Social Networks to Support Mothers, Fathers, or Guardians of Transgender Children and Adolescents: A Systematic Review"

_ijerph, 2022, doi:10.3390/ijerph19137941_

Round 1

Reviewer 1 Report

The subject of the work is interesting and its realization seems very attractive and pertinent, however in the manuscript important methodological deficits are appreciated.

The research question is poorly formulated, since we are not working on a review of experimental studies, it does not make sense to use the PICOS strategy, but instead the PECOS strategy should be used.

The process of searching for information is not sufficiently explicit

The data search procedure seems inadequate. It works with an initial volume of publications greater than 8000. This suggests an inadequate search process on which there is also insufficient information.

The flowchart indicates that 7,707 papers were excluded by individual review. How was this process carried out, how many people and how much time was invested in this task?

The results section is not developed. The tables should never replace the description of the data obtained

Author Response

To answer the first reviewer question, we opted for the PICo strategy as it is the most suitable for systematic reviews in qualitative researches–as indicated by reference number 14 from Joanna Briggs Institute. Contrary to what has been said, the PICo Strategy is not the same as PICO, nor PECO, because the qualitative investigation aims for phenomenons of interest: the social support present in the subjectiveness of experiences and speech of the population studied, and not observational or experimental analysis. Although a high numbers of studies were initially extracted from the bases, those originated from a targeted research strategy published in a protocol article that can be observed in reference 18. This review followed methodological rigor at all stages.

Reviewer 2 Report

Thank you for your work on this paper. This is an important and timely review to add to the literature. You did an excellent job compiling meaningful articles and findings., including the descriptions of the qualitative evaluation methods. There are several grammatical changes needed. I have noted some on the attached document. The conclusion needs editing to be more than a reiteration of the listed findings in the chart. There are a few formatting suggestions as well. Overall, this paper is valuable, important, and well written. I look forward to your revisions.

Author Response

Confusing sentences were rewritten, in addition we referenced Pauline Boss and inserted the definition of ambiguous loss. Necessary grammatical corrections were made, summarization and formatting errors were corrected.

Reviewer 3 Report

  1. Abstract. I don't really understand why mothers is a keywords.
  2. Intro. I don't understand the purpose of this sentence (lines 30-31) "Thus, there is a diversity of gender identity, which is independent of what is normalized by biological sex (“boy” or “girl”) or sexual orientation". Maybe a definition by an association or community could be beneficial. 
  3. Why is “trans” (line 30) between quotation marks? Is the authors questioning the term? 
  4. In addition is repeated twice in a sentence (lines 38-39).
  5. Intro. Intro can be more trans affirmative. When you read it, it provokes a sense that the authors want to be conservative and neutral. And conservative a neutral is not trans affirmative. 
  6. Figure 1 has some spacing errors. 
  7. Please use the word Table instead of chart. 
  8. Char 1 has some spacing issues too. Please be consistent with spaces. 
  9. Discussion. The is an error in the references "[23,-24, 26-27, 30, 33, 35]" the comma after the 23 should not be there

Author Response

The descriptor “mothers” was removed as recommended; in the introduction, the sentence (lines 30-31) was removed as requested without prejudice to the understanding of the paragraph. The quotation marks of the term “trans” were removed, as they were not necessary, terms repeated in the same sentence (lines 38-39) were removed. The authors added excerpts in the transaffirmative introduction (The analysis of scientific evidence can contribute to affirmative actions for trans visibility and endorse the criticality of the theme in the different social and political contexts around the world, mainly because those are vulnerable people whose identities and gender expressions are considered crimes in some countries. The political moment must be considered for an in-depth analysis of the genocide actions veiled in discriminatory speeches from powerful leaders that have repercussions on transphobia, social deaths and murders that are still recurrent even in so-called democratic countries.). In figure 1 the spacing errors have been corrected. The word graph was replaced by table; in the discussion the error in the reference was also corrected.

Reviewer 4 Report

This illuminating article reviews the research on the dynamics of primary social networks for the parents and caretakers of transgender youth.  On the basis of a very rigorous and thoughtfully explained selection procedure, the authors synthesize the main results of 21 journal articles, and analyze their methodological strengths and limitations.  The breadth of references identified and the charts presenting the studies’ core findings are both highly impressive.  This comprehensive, skillfully written review article can be of real use to researchers, advocates, and policymakers working around issues of transgender equity.  

I am excited about this article and believe that it can make a strong contribution to IJERPH.  That being said, I believe the manuscript would benefit from revision, specifically in the Discussion section.  I have 3 main suggestions, offered in the spirit of making this paper even stronger.  

1.  In the article’s current form, the Results section is quite strong and well-organized, but it is a little difficult to identify the key themes in the Discussion section.  I believe that this is primarily a matter of structure: without subheadings or language to cue the reader that a certain theme is central, we end up learning lots of interesting ideas from the studies, but not coming away with a clear sense of what the primary take-aways are.  I would encourage the authors to consider explicitly naming 4-5 main themes that emerged from their review and organizing the Discussion section on that basis.   

2.  I consider the geographic and historical range of the studies included in this article to be a real asset.  The studies range in publication date from 2009-2021, and are from the United States, Europe, and Oceania.  The review article emphasizes common themes across all the studies.  Yet, given that different countries provide vastly different legal and cultural contexts for raising trans youth, and given that conditions for trans youth have shifted dramatically, even just over the past 15 years, I would imagine that the primary social networks of caretakers for trans youth would be impacted by place and historical moment.  If there is simply more continuity than variation across all studies, that too would be quite interesting to know.  In any case, I would encourage the authors to explicitly address any impact that place and historical moment seem to make for the object of study (primary social networks for caretakers of trans youth) in the discussion. 

3.  Perhaps out of humility, the authors do not make a strong case for how their review article fills a gap in our knowledge, but I would encourage them to do so!  This article makes a significant contribution to our understanding of family dynamics and transgender experience.  There are, to my knowledge, no other review articles addressing this topic.  This review article is also particularly relevant in this political moment when caretakers’ efforts to support trans youth are being criminalized.  The authors approach to the topic is also innovative: scholarly and policy discussions about trans youth tend to focus on a single country, rather than thinking internationally.  I would encourage the authors to really own and be explicit about the importance and impact of their work.

Congratulations on an excellent article!

Author Response

In the discussion, we included sub-themes according to the content of the paragraphs, considering the main themes that emerged: Recognition of gender identity and the challenges faced; The existingties in the primary social networks; The lack of informational support and its impact. The description of the studies was rigorously detailed to cover all aspects that emerged in the studies, however, historical issues were not addressed or emphasized in the included studies. Added a paragraph on political moment and trans criminalization and relationship with support. The importance of the work was mentioned, especially an affirmative trans scientific literature with the active participation of trans researchers.

Round 2

Reviewer 1 Report

It would be interesting to indicate why the search not to start from such a high articles recovery number